Relationships between different components of intolerance of uncertainty and symptoms of obsessive–compulsive disorder: a network analysis

Ding XiaoBin 1
Zhao Ze 907964165@qq.com 1 2
Wang Jie 3
Chen Chen 1
Ding ShuChan 1
Gao JingYi 1
Deng Jun 1 4
Liu Dan 1 5
1 School of Psychology, The Northwest Normal University , Lanzhou , Gansu , China
2 Mental Health Education Centre, Nanchong Vocational and Technical College , Nanchong , Sichuan , China
3 Department of Teacher Education, Nanchong Vocational and Technical College , Nanchong , Sichuan , China
4 School of Architecture, Chengdu Jincheng College , Chengdu , Sichuan , China
5 School of Education, Tongren University , Tongren , Guizhou , China
Liuzza Marco Tullio
Electronic publication date: 2025 Jul 31
Publication date: 2025
Volume: 13
Electronic Location ID: e19791
Received 2024 Dec 19; Accepted 2025 Jul 3
Copyright: ©2025 Ding et al.
Copyright year: 2025
Copyright holder: Ding et al.
License: This is an open access article distributed under the terms of the Creative Commons Attribution License, which permits unrestricted use, distribution, reproduction and adaptation in any medium and for any purpose provided that it is properly attributed. For attribution, the original author(s), title, publication source (PeerJ) and either DOI or URL of the article must be cited.
License URL: https://creativecommons.org/licenses/by/4.0/

Keywords: Obsessive-compulsive disorder, Intolerance of uncertainty, Network analysis, Expected influence, Bridge expect influence

Funding: The Northwest Normal University Graduate Research Grant Program 2023KYZZ—B036 This work was funded by the Northwest Normal University Graduate Research Grant Program (2023KYZZ—B036). The funders had no role in study design, data collection and analysis, decision to publish, or preparation of the manuscript.

==============================
Background

Previous studies have shown that intolerance of uncertainty (IU) and obsessive–compulsive disorder (OCD) are closely interrelated. This reliance on scale totals to measure symptom severity obscures the distinctions and connections between different symptoms. In the present study, we explored the relationships between different components of IU and symptoms of OCD.

Methods

We recruited 1,616 participants and retained 1,529 pieces of valid data. Components of IU were measured by the Chinese version of the Intolerance of Uncertainty Scale-Short Form, and symptoms of OCD were measured by the Chinese version of the Obsessive-Compulsive Inventory-Revised. The present study employs network analysis to examine both core and bridging symptoms within the context of the IU and OCD networks.

Results

In the overall network, the nodes with the highest expected influence (EI) were OCD3 (“I get upset if things don’t work out”), IU6 (“I can’t stand being taken by surprise”), and OCD6 (“It’s hard for me to control my thoughts”). The nodes with the highest bridge expected influence (BEI) were OCD3 (“I get upset if things don’t work out”), OCD9 (“I get upset when people change my plans”), and IU12 (“I must get away from all uncertain situations”). Within the IU community, the strongest edge was between IU1 (“Unforeseen events upset me greatly”) and IU2 (“It frustrates me not having all the information I need”). Within the OCD community, the strongest edge was between OCD10 (“I force myself to repeat certain numbers”) and OCD11 (“Sometimes, I force myself to bathe or wash myself because I feel dirty”). The strongest edge connecting the IU and OCD communities was between IU10 (“When I am uncertain I can’t function very well”) and OCD6 (“It’s hard for me to control my thoughts”). No significant gender differences were found in the network structure.

Conclusions

This study revealed specific component–symptom patterns between different facets of intolerance of uncertainty (IU) and various obsessive-compulsive symptoms. Understanding how distinct components of IU—an assumed risk factor—relate to specific OCD symptoms may inform targeted prevention and intervention strategies. For example, interventions aimed at OCD3, IU6, OCD9, and IU12 may effectively reduce the severity of obsessive-compulsive symptoms among Chinese university students, enhance their ability to cope with uncertainty, and help disrupt the reciprocal influence between IU components and OCD symptoms.

Introduction

Obsessive-compulsive disorder (OCD) has been identified by the World Health Organization as one of the top ten causes of global health burden (Baxter et al., 2014). In China, the lifetime prevalence of OCD is 2.3%, with a mean age of onset of 19.5 years (Cruz et al., 2022). A recent meta-analysis reported that the lifetime prevalence is 1.5% for females and 1.0% for males, and that younger individuals are 1.4 times more likely to develop OCD than older adults (Fawcett, Power & Fawcett, 2020). OCD is characterized by the recurrent and persistent intrusion of thoughts, impulses, or images that cause significant anxiety or distress in most individuals, manifesting in two core dimensions: obsessions and compulsions (Abramowitz, Taylor & McKay, 2009; Kuty-Pachecka, 2021). The level of social functional impairment caused by OCD is comparable to that seen in schizophrenia; however, its underlying mechanisms remain unclear (Gao et al., 2022).

Brakoulias et al. (2014) proposed that cognitive factors play a crucial role in the development and maintenance of OCD. While both individuals with and without OCD may experience intrusive thoughts, impulses, or images, non-OCD individuals are generally able to ignore or dismiss them, whereas individuals with OCD perceive them as highly distressing (Purdon, 2023). According to cognitive models of OCD, patients interpret these intrusions in a negative and threatening manner, which leads to a series of defensive behaviors. These behaviors, in turn, reinforce irrational beliefs, creating a vicious cycle of obsession and compulsion (Gillan & Robbins, 2014). This view is consistent with that of Lazarov et al. (2012), who argued that obsessive doubt plays a central role in the development and persistence of OCD symptoms, and that the pursuit of certainty drives many of the behaviors seen in the disorder. The tendency to overestimate the threat of intrusive content and the excessive need for certainty closely align with a cognitive vulnerability known as intolerance of uncertainty (IU), which has been identified as a potential triggering factor for compulsive behaviors (Gillett et al., 2018; Pinciotti, Riemann & Abramowitz, 2021; Jalal, Chamberlain & Sahakian, 2023).

IU was originally identified in the context of generalized anxiety disorder (GAD) (Bottesi et al., 2020), and has since been recognized as a broad transdiagnostic predictor across various forms of psychopathology (Knowles & Olatunji, 2023). To date, considerable debate has surrounded the definition of IU, with different conceptualizations carrying distinct implications and associations within psychopathology (Starcevic & Berle, 2006). Freeston et al. (1994) defined IU as “a set of cognitive, emotional, and behavioral reactions to uncertainty in everyday life”. Sookman & Pinard (1995) described IU as “difficulty coping with ambiguity, novelty, and unpredictable change” Ladouceur, Gosselin & Dugas (2000) characterized IU as “a tendency to react negatively to uncertain events or situations, regardless of their likelihood or potential consequences”. Dugas, Gosselin & Ladouceur (2001) conceptualized IU as “a disposition to consider the occurrence of negative events as unacceptable, no matter how low the probability”. A widely accepted definition describes IU as “a dispositional incapacity to endure the aversive response triggered by the perceived absence of salient, key, or sufficient information, and sustained by the associated perception of uncertainty” (Carleton, 2016).

The relationship between IU and emotional disorders such as obsessive-compulsive disorder (OCD) has long been a subject of investigation (Gu et al., 2020). Boelen & Reijntjes (2009) found that, after controlling for shared symptom variance, the association between IU and OCD was stronger than its association with social anxiety, generalized anxiety disorder (GAD), or depression. Sarawgi, Oglesby & Cougle (2013) reported that IU predicted ordering, checking behaviors, and contamination fears in a non-clinical sample. Reuther et al. (2013) demonstrated that IU fully mediated the relationship between perfectionism and OCD severity. The most direct evidence comes from a study by Geok, Lee & Sündermann (2022), who manipulated feedback under conditions of high and low IU. They found that IU scores significantly decreased in the low-IU group following the intervention. Moreover, individuals in the low-IU group with higher baseline OCD symptoms exhibited greater reductions in OCD symptoms post-manipulation compared to those in the high-IU group. A review by Gillett et al. (2018) further emphasized that IU, as a transdiagnostic cognitive vulnerability, makes a significant contribution to the shared phenomenology of OCD and GAD, and may represent a promising target for intervention.

It is worth noting that the afore-mentioned studies primarily assessed symptom severity using total scale scores, a practice that may obscure the distinctions and interrelations among individual symptoms (Fried & Nesse, 2015; Fried et al., 2014). According to the network theory of mental disorders (Borsboom, 2017), psychopathology arises from the dynamic interactions between symptoms. The presence of one symptom may increase the likelihood of other related symptoms, and when tightly interconnected symptom clusters mutually reinforce each other, mental disorders can emerge and lead to the activation of a broader symptom network. Therefore, identifying central and bridge symptoms within such networks through network analysis holds significant practical value for developing targeted interventions. Previous studies have explored the network structure of both OCD and IU independently. For instance, Bottesi et al. (2020) applied network analysis to examine the internal structure of IU and found that the most central nodes were IU6 and IU7. Their results indicated that IU consists of three communities: negative beliefs, behaviors, and emotions centered around uncertainty. Olatunji et al. (2019) identified negative appraisals of intrusive thoughts as the most central symptom in the OCD network. Similarly, Cervin et al. (2022) used network analysis to investigate the symptom structure of clinical OCD patients and identified incompleteness and disturbing thoughts as core symptoms. In addition, a growing number of studies have investigated the comorbidity network structures involving IU and generalized anxiety disorder (Ren et al., 2021), OCD and depressive symptoms (Ma et al., 2023; Jones et al., 2018), eating disorders and OCD (Vanzhula, Kinkel-Ram & Levinson, 2021), as well as body dysmorphic disorder and OCD (Song et al., 2025).

However, despite substantial evidence supporting a close relationship between IU and OCD, no study to date has directly examined the component–symptom interaction patterns between IU and OCD within a sample of Chinese university students. Therefore, the present study integrates IU components and OCD symptoms into a unified network.

The objectives of this study are threefold:

(1) To identify the core symptoms within the IU–OCD network using centrality indices;

(2) To examine which symptoms serve as bridges that maintain the connectivity of the overall network using bridge centrality indices;

(3) To explore whether the network structure differs by gender.

Based on prior literature, we hypothesized that:

(1) OCD symptoms would be strongly associated with negative beliefs about uncertainty;

(2) Obsessions would emerge as central symptoms within the OCD network.

Method

Ethics declarations

This study was approved by the Human Research Ethics Committee of Northwest Normal University (No. 2023187). All participants were instructed to carefully read the questionnaire instructions before beginning the survey. In the instructions, participants were clearly informed that the survey was anonymous, that no personal information would be disclosed, and that proceeding with the questionnaire would be considered as providing informed consent for participation.

Participants

This study employed a cross-sectional design, and data were collected via Wenjuanxing, a widely used online survey platform in China. To ensure the representativeness of the sample, we adopted a simple random sampling method. Using a random number generator, we randomly selected four eligible higher education institutions from western China, including both vocational colleges and regular undergraduate universities. The target population of this study was Chinese university students, a group with high levels of digital literacy and internet usage, often referred to as “digital natives”. Therefore, the risk of exclusion due to limited internet access or technical skills is minimal. According to the 2023 CNNIC report, individuals aged 20–29 account for 24.5% of all internet users in China, further supporting the representativeness of our sample. Participants were invited to complete the questionnaire through institutional email systems, class chat groups, and online announcements.

Prior to beginning the questionnaire, all participants were required to carefully read an online informed consent form embedded at the start of the survey. This form clearly stated the purpose of the study, emphasized the voluntary nature of participation, and provided details on anonymity and confidentiality. Specifically, participants were informed that no personally identifiable information would be collected, and all responses would be used exclusively for academic research. Proceeding with the survey was considered as providing informed consent. To balance data completeness and ethical considerations, only essential demographic variables and core variables relevant to network analysis were set as mandatory in the Wenjuanxing platform. All other items, particularly those that might be sensitive or emotionally triggering, were optional and could be skipped at the participant’s discretion. The current network analysis aimed to estimate 30 nodes and 325 edges, which required a sufficiently large sample size. The entire sampling and recruitment process was carefully documented to ensure transparency and reproducibility. According to the recommendation by Fried & Cramer (2017), a minimum of three participants per estimated parameter is required, suggesting that at least 1,095 participants were needed for stable network estimation. A total of 1,616 participants were recruited for this study. Among them, 58 participants did not provide valid age information, and 22 participants were under the age of 18. Their data were excluded from the final analysis. The final sample consisted of 1,529 participants (60.1% female; Mage = 19.62, SD = 1.38; age range = 18–25 years).

Measures

Intolerance of uncertainty

The Chinese version of the Intolerance of Uncertainty Scale–Short Form (C-IUS-12), revised by Zhang et al. (2017), was used to assess different components of IU. The scale consists of 12 items rated on a 5-point Likert scale ranging from 1 (“Not at all characteristic of me”) to 5 (“Entirely characteristic of me”). The Chinese version of the total scale demonstrated good internal consistency, with a Cronbach’s alpha of 0.907. In the present study, the scale also showed excellent internal consistency (Cronbach’s α = 0.92).

Obsessive-compulsive symptoms

The Chinese version of the Obsessive-Compulsive Inventory-Revised (OCI-R), revised by Tang et al. (2011), was used in this study (Cronbach’s α = 0.895). The OCI-R is one of the most widely used instruments for assessing obsessive-compulsive symptoms (Abramovitch, Abramowitz & McKay, 2021). It comprises 18 items across six dimensions: Washing, Checking/Doubting, Obsessing, Mental Neutralizing, Ordering, and Hoarding. Items are rated on a 5-point scale from 0 (“Not at all distressing”) to 4 (“Extremely distressing”), with higher scores indicating greater levels of distress. In the current study, the internal consistency of the Chinese version of the OCI-R was excellent (Cronbach’s α = 0.94).

Data analysis

Following the recommendations of Epskamp & Fried (2018), we constructed a Spearman correlation matrix to estimate the network structure. The network was constructed and visualized using the R package qgraph. In the visualized network, blue edges indicate positive associations and red edges indicate negative associations. To avoid topological overlap between items in the network (McNally, 2021), we used the goldbricker function from the networktools R package to assess item redundancy. Based on the guidelines proposed by Levinson et al. (2018), we set the significant proportion for inclusion at 0.25 and the p-value threshold at 0.01. The network was estimated using a graphical Gaussian model (GGM), regularized with the Graphical Least Absolute Shrinkage and Selection Operator (GLASSO) and the Extended Bayesian Information Criterion (EBIC). This approach reduces false positives and yields a sparse, interpretable network structure (Bai et al., 2022). The tuning parameter was set to 0.5 to strike a balance between sensitivity and specificity in edge selection (Ren et al., 2021). Expected influence (EI) of each node was computed using the qgraph package, while bridge expected influence (BEI) was calculated using the networktools package (Jones, Ma & McNally, 2021). Node predictability was assessed using the mgm package. Network robustness was evaluated using the bootnet R package. A nonparametric bootstrap procedure with 1,000 samples was employed to estimate 95% confidence intervals for edge weights and to test for significant differences between edge weights (Feng et al., 2022; Yuqing et al., 2020). The stability of centrality indices (EI and BEI) was assessed using a case-dropping bootstrap procedure with 1,000 iterations. Prior research suggests that the correlation stability coefficient (CS-coefficient) should ideally exceed 0.50 and should not fall below 0.25 for the results to be considered interpretable (Epskamp, Borsboom & Fried, 2018).

Network comparisons between gender groups were conducted using the Network Comparison Test (NCT) package (Van Borkulo et al., 2023). We performed 1,000 permutations to compare the distribution of edge weights between male and female networks. To control for multiple comparisons, the Holm-Bonferroni correction was applied, and differences in edge strength between the two gender-specific networks were evaluated accordingly.

Results

Item selection

Table 1 presents all items included in the final network analysis. A topological overlap check identified four pairs of overlapping items: IU10 & IU9, OCD5 & OCD4, OCD18 & OCD7, and OCD17 & OCD16, indicating substantial structural similarity between the items within each pair. To avoid redundancy, one item from each pair was considered for removal. We conducted correlation difference tests for IU10 & IU9 with all OCD items, and similarly for OCD5 & OCD4, OCD18 & OCD7, and OCD17 & OCD16 with all IU items, using zero as the test value. The results were as follows: tr(IU9−IU10) = −0.032, p = 0.98; tr(OCD4−OCD5) = −4.801, p < 0.01; tr(OCD7−OCD18) =1.72, p = 0.11; and tr(OCD16−OCD17) =2.93, p < 0.05. These results indicate that, compared to OCD4, OCD5 exhibited significantly stronger correlations with all IU items; similarly, compared to OCD17, OCD16 showed stronger correlations with all IU items. Although the differences between IU9 and IU10, and between OCD7 and OCD18 were not statistically significant, IU10 demonstrated slightly stronger correlations with all OCD items than IU9, and OCD7 exhibited slightly stronger correlations with all IU items than OCD18. Based on these findings, IU9, OCD4, OCD17, and OCD18 were excluded from the final network, resulting in a total of 26 items retained for analysis.

Table 1 Abbreviation, means, standard deviation (SD), skewness, and kurtosis of each variable in the present network.

Items	Abbreviation	M(SD)	Skewness	Kurtosis	
Components of intolerance of uncertainty					
IU 1 (Unforeseen events upset me greatly)	IU1	1.94 (0.99)	1.22	1.29	
IU 2 (It frustrates me not having all the information I need)	IU2	1.79 (0.95)	1.44	2.05	
IU 3 (One should always look ahead so as to avoid surprises)	IU3	1.84 (1.02)	1.35	1.41	
IU 4 (A small, unforeseen event can spoil everything, even with the best of planning)	IU4	2.29 (1.13)	0.75	−0.11	
IU 5 (I always want to know what the future has in store for me)	IU5	1.82 (0.98)	1.42	1.81	
IU 6 (I can’t stand being taken by surprise)	IU6	1.83 (0.94)	1.38	2.00	
IU 7 (I should be able to organize everything in advance)	IU7	1.95 (0.98)	1.14	1.17	
IU 8 (Uncertainty keeps me from living a full life)	IU8	2.38 (1.20)	0.71	−0.38	
IU 10 (When I am uncertain I can’t function very well)	IU10	1.73 (0.93)	1.42	1.75	
IU 11 (The smallest doubt can stop me from acting)	IU11	2.42 (1.09)	0.61	−0.19	
IU 12 (I must get away from all uncertain situations)	IU12	1.93 (0.97)	1.14	1.13	
Symptoms of obsessive-compulsive disorder					
OCD1 (I have been hoarding a lot of things)	OCD1	1.94 (0.98)	1.03	0.69	
OCD2 (I always double-check, whether I need to or not)	OCD2	1.91 (0.95)	0.99	0.59	
OCD3 (I get upset if things don’t work out)	OCD3	2.04 (1.00)	0.87	0.24	
OCD5 (If I know something’s been touched by a stranger I can’t bring myself to touch it again)	OCD5	1.57 (0.83)	1.58	2.44	
OCD6 (It’s hard for me to control my thoughts)	OCD6	1.74 (0.96)	1.42	1.70	
OCD7 (I collect things that I don’t need)	OCD7	1.64 (0.87)	1.55	2.43	
OCD8 (I double-checked the doors, windows, drawers)	OCD8	1.66 (0.90)	1.48	2.02	
OCD9 (I get upset when people change my plans)	OCD9	1.98 (0.98)	1.02	0.73	
OCD10 (I forced myself to repeat certain numbers)	OCD10	1.43 (0.78)	2.20	5.22	
OCD11 (Sometimes, I force myself to bathe or wash myself because I feel dirty)	OCD11	1.54 (0.85)	1.76	3.04	
OCD12 (I get upset when I think the opposite of what I want)	OCD12	1.80 (0.92)	1.23	1.41	
OCD13 (I don’t throw things away because I’m afraid I’ll need them later on)	OCD13	2.23 (1.08)	0.65	−0.38	
OCD14 (I double-check the gas, tap and light switches after switching them off)	OCD14	1.98 (1.01)	0.93	0.27	
OCD15 (I need things in a certain order)	OCD15	1.96 (0.97)	0.97	0.56	
OCD16 (I think there are good and bad numbers)	OCD16	1.71 (0.92)	1.32	1.38	

Descriptive statistics

The means, standard deviations, kurtosis, skewness and abbreviations of the variables used in the present study are listed in Table 1.

Network structure

As shown in Fig. 1, the network exhibited the following characteristics: among the 325 possible edges, 181 (55.69%) were non-zero, with an average edge weight of 0.04. Node IU6 showed the highest predictability (0.69), indicating that 69% of its variance could be explained by its neighboring nodes. The strongest edge connecting the IU and OCD communities was observed between IU10 (“When I am uncertain I can’t function very well”) and OCD6 (“It’s hard for me to control my thoughts”), with a weight of 0.11. Within the OCD community, the strongest edge was between OCD10 (“I forced myself to repeat certain numbers”) and OCD11 (“Sometimes, I force myself to bathe or wash myself because I feel dirty”), with a weight of 0.31; within the IU community, the strongest edge was between IU1 (“Unforeseen events upset me greatly”) and IU2 (“It frustrates me not having all the information I need”), also with a weight of 0.29.

Figure 1 Network structure of IU and OCD in university students.

Note: The size of the correlation was reflected in the thickness of the edge. The text of intolerance of uncertainty and OCD can be seen in Table 1.

As shown in Fig. 2, the Expected Influence (EI) values of nodes in the overall network structure are presented. OCD3 (“I get upset if things don’t work out”) had the highest EI, followed by IU6 (“I can’t stand being taken by surprise”) and OCD6 (“It’s hard for me to control my thoughts”). The bootstrapped 95% confidence intervals were relatively narrow, indicating that the edge weights in the IU–OCD network were estimated with good precision (see Fig. S1). Results from the bootstrapped difference test for edge weights are presented in Fig. S2, showing that edges with higher weights differed significantly from those with lower weights. The bootstrapped difference test for node EI further indicated that nodes with high EI values significantly differed from most other nodes, whereas nodes with low EI did not show significant differences (see Fig. S3).

Figure 2 Centrality plot depicting the expected influence (z-score) of each variable chosen in the network.

As shown in Fig. 3, the node with the highest Bridge Expected Influence (BEI) was OCD3 (“I get upset if things don’t work out”), followed by OCD9 (“I get upset when people change my plans”). Within the IU community, IU12 (“I must get away from all uncertain situations”) showed the highest BEI, suggesting that IU12 has the strongest connections to obsessive-compulsive symptoms among all IU items. Results from the bootstrapped difference test for BEI indicated that nodes with high BEI significantly differed from most other nodes, whereas those with low BEI did not show significant differences (see Fig. S4).

Figure 3 Centrality plot depicting the bridge expected influence (z-score) of each variable chosen in the network.

The stability of node expected influence (EI) and bridge expected influence (BEI) was assessed using a case-dropping bootstrap procedure. The correlation stability (CS) coefficients for both EI and BEI were 0.75, exceeding the recommended threshold of 0.5, indicating that the centrality indices were sufficiently stable (see Fig. 4).

Figure 4 Stability of node expected influences and bridge expect influence.

Network comparison

Due to the substantial difference in sample size between male participants (n = 610) and female participants (n = 919), direct comparison might lead to biased results (Van Borkulo et al., 2023). Therefore, we randomly selected a subsample of female participants equal in size to the male group and constructed gender-specific networks (see Fig. S5). The significance levels (p-values) were adjusted using the Bonferroni–Holm correction. Results showed no significant difference in network structure invariance between males and females (M = 0.22, p = 0.053; see Fig. S6), and no significant difference in global strength (male network strength = 12.90, female network strength = 12.25; S = 0.65, p = 0.09; see Fig. S6).

Discussion

Through network analysis, we aimed to uncover the interaction patterns between different components of IU and obsessive-compulsive symptoms. Synthesizing the findings above, we observed that in the current IU-OCD network, the nodes with the highest EI were OCD3, IU6, and OCD6, while the nodes with the highest BEI were OCD3, OCD9, and IU12. The strongest intra-community edge in the IU cluster was found between IU1 and IU2, and in the OCD cluster between OCD10 and OCD11. The strongest inter-community edge between IU and OCD was identified between IU10 and OCD6. Consistent with previous research (Ren et al., 2021; Feng et al., 2022), the estimated network revealed that within-community connections tended to be denser and stronger than between-community connections.

The node OCD3 (“I get upset if things don’t work out”) showed the highest EI, indicating that this symptom is the most central node in the current network model. OCD3 belongs to the Ordering dimension of the OCI-R, which contrasts with findings from previous OCD network studies based on dimensional structures (Olatunji et al., 2019; Song et al., 2025). However, this result aligns with the findings of Wang et al. (2023), who identified Ordering as the most central dimension in patient networks. This finding supports cognitive models of OCD, which suggest that dysfunctional beliefs lead to obsessive anxiety, and that engaging in compulsive behaviors serves to alleviate intrusive thoughts and associated distress (Gillan & Robbins, 2014). The nodes IU6 (“I can’t stand being taken by surprise”) and OCD6 (“It’s hard for me to control my thoughts”) also demonstrated high EI values. OCD6 belongs to the Obsessing dimension of the OCI-R, further supporting our Hypothesis 2. Consistent with previous research (Olatunji et al., 2019; Cervin et al., 2022), obsessive thoughts emerged as the most central symptoms in OCD networks. In network structures derived from children and adolescent samples, doubt has been found to play a key central role across all symptom dimensions (Cervin et al., 2020). Another network analysis based on OCD patients revealed that the most central symptoms were related to interference caused by compulsions and obsessive thoughts (Kim et al., 2023). According to the cognitive model of OCD, the concept of thought–action fusion (TAF) suggests that the disorder is rooted in the belief that merely thinking about a negative event increases the likelihood of its occurrence (Reynolds & Reeves, 2008). From this perspective, compulsive behaviors function to reduce uncertainty-induced arousal by enhancing perceived control over future outcomes (Greco & Roger, 2003). The identification of IU6 as a central node is also supported by previous research; for example, Bottesi et al. (2020) found IU6 to be the most central item in both undergraduate and community samples. This finding suggests that discomfort with uncertainty may play a particularly important role in the development of IU. All identified central nodes and components exhibited high predictability values, accounting for 55% of the variability in average node predictability across the network. Therefore, the central symptoms and components identified in this study may play a crucial role in the development and maintenance of IU and OCD in university students, and warrant further investigation in both experimental and clinical contexts (Fried et al., 2017).

The node OCD3 (“I get upset if things don’t work out”) showed the highest BEI, indicating that it had the strongest connections with IU components, followed by OCD9 (“I get upset when people change my plans”). This suggests that these two symptoms are closely related to various aspects of intolerance of uncertainty. Notably, both OCD3 and OCD9 belong to the Ordering dimension of the OCI-R, further underscoring the crucial role that the ordering dimension plays in maintaining the overall structure of the network. Previous research has also shown that the ordering dimension has a unique association with depressive symptoms (Cervin et al., 2020b). Traditional learning-based psychological models of OCD suggest that compulsive behaviors are primarily reinforced by the avoidance of harm. However, ordering-related symptoms appear to reflect a need for the environment to “feel right”. These symptoms represent a distinct subtype of OCD, in which compulsions are driven not by threat avoidance per se, but by the urge to eliminate unpleasant “not-just-right” feelings (Fineberg et al., 2018). Within the IU network, IU12 (“I must get away from all uncertain situations”) demonstrated the highest BEI among all IU items, indicating that it is more strongly linked to obsessive-compulsive symptoms than other IU components. From a network perspective, targeting IU12 may be more effective in reducing OCD symptoms than interventions aimed at other aspects of IU. Psychopathological models of IU propose that, in uncertain situations, anxious individuals tend to overestimate the likelihood or severity of potential threats (Knowles & Olatunji, 2023). In fact, individuals with OCD frequently exhibit threat overestimation, particularly in response to threats that are personally relevant (Taylor et al., 2010).

In the IU network, the strongest edge was found between IU2 and IU1, which is consistent with previous findings (Liu et al., 2022) and aligns with results from earlier IU network studies (Ren et al., 2021). Both IU1 and IU2 belong to the emotional response community to uncertainty within the IU network (Bottesi et al., 2020), further supporting the notion that feelings of agitation and emotional distress are closely tied to uncertainty (Liu et al., 2022). In the OCD network, the strongest edge was observed between OCD10 and OCD11, which belong to the Washing and Mental Neutralizing dimensions of the OCI-R, respectively. Wang et al. (2023) found that even healthy individuals tend to score relatively high on the Contamination/Cleaning dimension of the OCI-R, and that this may be associated with childhood trauma. Therefore, the strong association between repetitive counting and compulsive washing observed in the present study may reflect a precursor response that could later develop into clinical OCD. Longitudinal research is needed to further explore this developmental trajectory. The strongest cross-community edge between IU and OCD was found between IU10 and OCD6. Contrary to our hypothesis, IU10 belongs to the behavioral response community of the IU network (Bottesi et al., 2020). The strong connection between behavioral freezing in response to uncertainty (IU10) and intrusive thoughts that feel uncontrollable (OCD6) is in line with findings by Pinciotti, Riemann & Abramowitz (2021), who suggested that individuals with obsessive thoughts may experience behavioral paralysis when confronted with uncertainty (Keune et al., 2012). There is also consistent evidence that changes in IU are closely linked to changes in OCD symptoms (Knowles & Olatunji, 2023). Taken together, these findings suggest that targeting OCD3, IU6, OCD9, and IU12 in interventions may be effective in reducing the overall connectivity and intensity of the symptom network, thereby disrupting the mutual reinforcement between symptoms. These results offer practical implications for enhancing uncertainty tolerance and developing targeted prevention and intervention strategies for obsessive-compulsive symptoms among Chinese university students. In addition, we found no significant differences between the male and female networks, which may be due to the cross-gender stability of the IU network structure (Bottesi et al., 2020). More importantly, the majority of IU is composed of time-invariant (TI) components, which show the strongest associations with OCD (Knowles et al., 2022), making it difficult to detect gender differences within the current cross-sectional design.

To the best of our knowledge, this is the first study to explore the network structure of IU and obsessive-compulsive symptoms among Chinese university students. However, several limitations should be acknowledged. First, our sample consisted exclusively of university students, and we examined a range of IU components and OCD symptoms spanning from subclinical to potentially clinical levels. This may limit the generalizability of our findings. The resulting network structure and related indicators (e.g., expected influence and bridge expected influence) may differ in clinical populations. Second, the study was based on cross-sectional data, which prevents us from drawing conclusions about the directionality of relationships among variables. Future studies should employ longitudinal designs to examine potential causal relationships between IU and OCD symptoms. Third, both IU and OCD were assessed using self-report questionnaires, which may introduce recall bias and limit the accuracy of symptom reporting.

Conclusion

The present study contributes to a deeper understanding of the relationship between intolerance of uncertainty and obsessive-compulsive symptoms by employing a network analysis approach to map the connections between specific IU components and distinct OCD symptoms. Our findings reveal that certain behavioral responses to uncertainty are closely linked to core obsessive-compulsive symptoms, emphasizing the importance of bridging mechanisms between the two constructs. These insights not only enrich the theoretical framework of IU–OCD comorbidity but also offer practical implications for designing more targeted and component-specific interventions aimed at reducing obsessive-compulsive symptoms, particularly in university student populations.

Supplemental Information

Supplemental Information 1 Network analysis

Supplemental Information 2 Supplementary material

Additional Information and Declarations

Competing Interests

Author Contributions

Human Ethics

Data Availability

The authors declare there are no competing interests.

XiaoBin Ding conceived and designed the experiments, analyzed the data, prepared figures and/or tables, authored or reviewed drafts of the article, and approved the final draft.

Ze Zhao conceived and designed the experiments, performed the experiments, analyzed the data, prepared figures and/or tables, authored or reviewed drafts of the article, and approved the final draft.

Jie Wang conceived and designed the experiments, performed the experiments, authored or reviewed drafts of the article, and approved the final draft.

Chen Chen conceived and designed the experiments, performed the experiments, authored or reviewed drafts of the article, and approved the final draft.

ShuChan Ding conceived and designed the experiments, performed the experiments, authored or reviewed drafts of the article, and approved the final draft.

JingYi Gao conceived and designed the experiments, performed the experiments, authored or reviewed drafts of the article, and approved the final draft.

Jun Deng conceived and designed the experiments, performed the experiments, authored or reviewed drafts of the article, and approved the final draft.

Dan Liu conceived and designed the experiments, performed the experiments, authored or reviewed drafts of the article, and approved the final draft.

The following information was supplied relating to ethical approvals (i.e., approving body and any reference numbers):

The Human Research Ethics Committee of Northwest Normal University approved this study (2023187).

The following information was supplied regarding data availability:

The raw measurements are available in the Supplemental File.

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
