# Peer review of "Relationships between different components of intolerance of uncertainty and symptoms of obsessive–compulsive disorder: a network analysis"

_PeerJ, doi:10.7717/peerj.19791_

## Round 0.1 · original submission · Major Revisions

After careful consideration of the reviewers’ comments and a thorough evaluation of your work, we have reached a decision. Although your study addresses a timely and relevant topic in the field, substantial revisions are required before the manuscript can be considered for publication.

All three reviewers raised important concerns regarding conceptual clarity, methodological rigor, and ethical procedures. In particular, issues were noted regarding the theoretical framing of the constructs, the consistency between levels of analysis, the sampling and consent process, and the interpretation and presentation of the results. For these reasons, I am requesting a major revision.

We believe that your study has the potential to make a meaningful contribution, provided that the issues outlined in the reviews are carefully addressed. Please find below a summary of the key points raised by the reviewers, which we ask you to consider in preparing your revised manuscript.

Reviewer #1 suggests checking for item redundancy (topological overlap).
They also recommend dropping the flow analysis, as it is based on subscale-level constructs, while the network analysis is conducted at the item level. This mismatch in levels of analysis is also criticized by Reviewer #2, who finds the interpretation potentially misleading.

Reviewer # 2 also raises an important ethical issue questioning whether informed consent was obtained from underage participants. More details on the collection of informed consent are raised by Reviewer #3 as well. Additional concerns about how informed consent was obtained—particularly in the context of an online survey—are also raised by Reviewer #3.
Reviewer #2 questions not only the scientific impact of the investigation but also the theoretical grounding of the study. I suggest strengthening your arguments to clarify the rationale for the study and better justify its potential contribution to the literature.

I strongly suggest paying close attention to the criticism expressed by Reviewer #2 regarding the way you introduced the construct and its measures, as it seems that your introduction overlooks important parts of the literature.
Reviewer #2 also points to a lack of clarity in the presentation and interpretation of results and recommends a deeper explanation of the implications that can be drawn from the core network analyses (e.g., Figure S6).

Reviewer #3 requests more details about your sampling procedure and raises concerns about the generalizability of the findings, especially considering how participants were recruited and how the sample size was determined

**Language Note:** The review process has identified that the English language must be improved. PeerJ can provide language editing services - please contact us at [email protected] for pricing (be sure to provide your manuscript number and title). Alternatively, you should make your own arrangements to improve the language quality and provide details in your response letter. – PeerJ Staff

Reviewer 1 ·

Basic reporting

I suggest rephrasing the sentence “Several studies have shown that individuals with OCD exhibit unique cognitive processing [4] 51 [5], and that one of the key cognitive processes is intolerance of uncertainty” because IU is not a unique feature of OCD. It is, in fact, a transdiagnostic trait.

The authors state that “the centrality index was used to examine the relative importance….” This is inaccurate, as estimating relative importance requires a different centrality measure.

Experimental design

It appears that the sample included minors (age range: 16-25). Was parental consent obtained for these participants? If not, this subgroup should be excluded from the analysis.

Did the authors check for topological overlap? This can easily be done using the goldbricker function from the networktools R package.

I strongly recommend removing the flow function analysis, as it is based on the two subscales of the IUS. Given the item-level focus of the analysis, incorporating subscales could be conceptually misleading and confusing for readers. If the authors wish to keep this analysis, they should evaluate the subscales in network terms using community algorithms.

Validity of the findings

The authors report a significant global strength difference between males and females but fail to discuss this finding in the discussion section.

·

Basic reporting

The topic presented by the authors is of interest in the field of applied clinical psychology. The relationship between Intolerance of Uncertainty and OCD has been widely studied (Knowles and Olatunji, 2023; Moore et al., 2023; Pinciotti et al., 2021). However, as the authors indicate, there is not much scientific research from a network analysis perspective, which could provide interesting new insights in the area.

In spite of this, after carefully reviewing the authors' proposal, I would suggest that the manuscript not be accepted for publication for the following reasons.

In general terms, writing lacks clarity, with long sentences that are difficult to follow (i.e., p.7, lines 56 to 61)

The conceptualisation provided in the introduction lacks clarity too, and does not really provide an appropriate scientific review of the current knowledge regarding Intolerance of Uncertainty (IU) and its relationship with OCD symptomatology. In this regard, for instance, Intolerance of Uncertainty is defined in the third paragraph on page 7 (lines 50-63). However, in spite of all the massive scientific knowledge hitherto regarding IU, authors only provide two brief sentences that describe, in general terms, what is understood by Intolerance of Uncertainty, referring to Carleton's (2012; 2016) works. Surprisingly, immediately after, authors start by commenting on how IU is measured, mentioning whether the IUS-27 and IUS-12 show a two or five-factor structure. Why is this relevant for the present research? Are authors going to dive into the factorial structure of the questionnaire? If not, then why focus on this? The authors end this paragraph by briefly defining prospective and inhibitory uncertainty. However, in the later analyses, they seem to focus on the IUS-12 items individually, and not in the two factors. Why are then prospective and inhibitory uncertainty of relevance in the introduction? This is even more shocking when considering that what the authors aim to do is a network analysis focusing on the influence of each IUS-12 item on each OCD measure item.

After presenting the IU, the authors come back to OCD in the next paragraph, to comment on the “obsessive doubt” (p. 7, line 64). Why has it not been previously explained or mentioned in the first or second paragraph, when the authors were defining OCD and its main characteristics?

On page 8, the authors present the Network analysis proposal for getting a better understanding of how some aspects of the IU might enhance OCD symptomatology. However, there is a lack of clarity in the explanation, and a stronger scientific foundation would be required at this point.
There is another major conceptual mistake that, from an epistemological perspective, questions the whole research. Authors repeatedly refer to “the different components of IU” (i.e., p. 8, line 95). The items of the IUS-12 do not represent IU components. According to the latest evidences (Bottesi et al., 2019), IU seems to fit better within a unifactorial conceptualisation, instead of the two-factors approach (being the five factors solution widely discarded by empirical evidence; Boelen et al., 2016; Carleton et al., 2010; Fetzner et al., 2013; Mahoney and McEvoy, 2012). Regardless of the factorial structure of the IUS-12 (and the implications for the structure of the IU as a construct), IU components would refer to, precisely, these factors that might lie underneath the general factor; items of a questionnaire are never components. This same criticism would apply to the OCD. According to DSM-5 (APA, 2013), OCD would be comprised of two “components”: obsessions and/or compulsions. The items of a specific measurement of OCD symptomatology would not capture any components of the disorder. Therefore, the whole framework of the research is biased from the beginning.

The study also lacks information about the general and specific goals of the research. It neither informs about the working hypotheses that are going to be tested in the network analyses. This renders it impossible to properly assess the validity and adequacy of the results.

These conceptual problems continue in the “Methods” section. For instance, on p. 8, line 110, the authors indicate that “Given that the mean age of onset of OCD is 19.5 years…”. However, they provide as a reference a 2005 work (that means, published 20 years ago) conducted with English-speaking participants, as a justification for choosing a college sample. This is not a valid justification for why this sample was chosen.

Experimental design

-

Validity of the findings

Regarding the results, it was really hard to follow the reasoning and the interpretation of the results offered by the authors. The text is full of information that is already available in table 1 and figures S2 to S7, and, while in the body of the manuscript, perhaps it hinders the reading rather than facilitating it. Authors also seem to move to and fro when explaining the results, from one figure to the next, and coming back again. The explanation provided for the results found in the network analysis is not clear. It is also surprising that figure S6, which represents the core of the research aim (the network analysis of the relationship between IU and OCD is not explained in more detail. Specifically, a deeper explanation of the results found in that network would be fundamental for answering the study goals.

In relation to how the figures S3 to S7 are displayed, they are also difficult to understand. Why are IUS-12 and OCD items not presented in order? Figure S3 is particularly challenging in its reading and understanding. The legend in axis X and Y is impossible to read and, therefore, the interpretation of the information is not possible.

Lastly, authors include in the Discussion section many relevant references to previous scientific findings that were not explained in the introduction section (the part of the article where all relevant previous findings and research must be summarised). Why were these references not presented and explained in the introduction?

Additional comments

I acknowledge the great potential of this research and the importance of the data collected in a large sample size (something that, in psychology, is always welcomed). However, in its current state, I would recommend that the article not be accepted. Nonetheless, I would encourage authors to run a full, comprehensive review of their manuscript and submit it again.

Reviewer 3 ·

Basic reporting

1. English needs to be improved so that an international audience can see your text clearly.
2. References provide sufficient field background/context.
3. Figures are relevant to the content of the article. However, Figure 4 struggles with visual detail. They could be a bit clearer.

Experimental design

1. Research objectives are well defined, relevant, and meaningful.
2. There is no information on sample sizes, generalizability and whether it is sufficient for permanence tests. How was the number of students included in the study determined?
• How were the students selected?
• Were all students of the university reached? Were those who responded included in the study?
• How many students are actively attending the university? Was the study delivered to all of them?
3. Conducting the survey on an online platform may bring some limitations. For example, people with low digital literacy levels or those without internet access could not participate in this study. In addition, there may be doubts about anonymity and confidentiality in online surveys, and these responses may be incorrect.
4. It is stated that verbal consent was obtained from the participants; how was this done via WeChat? It is not stated whether sufficient information was given about the purpose of the study, whether the participation of the volunteers was ensured, or how the confidentiality guarantees were presented to the participants. In ethical terms, such information should be included.
5. Mandatory responses may cause some examples to have difficulty in giving correct answers, because some may have difficulty answering some questions or may give insincere answers. This situation ensures that the data is negatively affected. In addition, mandatory answers, relationships remain unanswered, and freedom is restricted for ethical reasons, so that it is personal.
6. Cronbach's alpha values of the scales revealed in the studies of Zhang et al 2017 and Abramovitch et al 2020 should be given.

Validity of the findings

The conclusion section should be detailed regarding the focal findings of the study.
Figure 4 is difficult to understand visually. It can be clarified a little more.

Additional comments

Providing an Innovative Approach to Psychopathology and Developing New Intervention Methods can be considered as the strengths of the study. However, the lack of information on sample size and sample selection, and the fact that it was conducted via WeChat, are considered the weaknesses of the study.

---

## Round 0.2 · Minor Revisions

After reading the two reviews, I consider the manuscript suitable for publication pending the minor revision requested by Reviewer #1

Reviewer 1 ·

Basic reporting

When assessing topological overlap, the authors chose to remove one item from each overlapping pair. Could the authors clarify why they opted for removal rather than merging the two nodes using a method such as PCA? Merging could have allowed for the retention of shared information. Additionally, the criteria used to determine which item to remove are not specified and should be briefly explained.

Among the three stated objectives of the study, the third was to explore gender differences. Although the authors report a null result, this aspect is not discussed further. While interpreting null findings can be challenging, given that this was one of the main aims of the study, it would be appropriate to briefly address this result in the Discussion section.

Experimental design

NA

Validity of the findings

NA

Additional comments

NA

·

Basic reporting

After reading the reviewed version of the manuscript submitted by authors, it is my impression that all my previous comments and the issues I highlighted have been favourably addressed and adequately solved. Therefore, my suggestion to the journal editor(s) is to accept the manuscript for its publication.

Experimental design

No comment

Validity of the findings

Authors have undoubtedly made a great effort in reviewing and improving both the quality of the analyses and the clarity of the results reporting.

Additional comments

No comment.

---

## Round 0.3 · accepted · Accept

I am glad to inform the authors that, after looking at their responses, I deem the manuscript suitable for publication on PeerJ.